# Effect of Selenium Nanoparticles and/or Bee Venom against STZ-Induced Diabetic Cardiomyopathy and Nephropathy

**DOI:** 10.3390/metabo13030400

**Published:** 2023-03-08

**Authors:** Mona M. Lotfy, Mohamed F. Dowidar, Haytham A. Ali, Wael A. M. Ghonimi, Ammar AL-Farga, Amany I. Ahmed

**Affiliations:** 1Biochemistry Departments, Faculty of Vet. Med., Zagazig University, Zagazig 44519, Egypt; 2Department of Biochemistry, College of science, University of Jeddah, Jeddah 23218, Saudi Arabia; 3Department of Histology and Cytology, Faculty of Vet. Med., Zagazig University, Zagazig 44519, Egypt

**Keywords:** SeNPs, bee venom, diabetic cardiomyopathy, diabetic nephropathy, miRNA 328, miRNA 21

## Abstract

The main purpose of our study was to examine the role of selenium nanoparticles (SeNPs) and/or bee venom (BV) in ameliorating diabetic cardiomyopathy (DCM) and nephropathy (DN) at the biochemical, histopathological and molecular levels. Fifty male albino rats were used in this experiment, divided into five groups: control, Streptozocin (STZ) diabetic, STZ-diabetic treated with SeNPs, STZ-diabetic treated with BV, and STZ-diabetic treated with SeNPs and BV. Biochemically, STZ injection resulted in a significant increase in serum glucose, BUN, creatinine, CRP, CK-MB, AST, LDH and cardiac troponins with a significant decrease in the serum insulin and albumin concentrations. Histopathologically, STZ injection resulted in diabetes, as revealed by glomerulonephritis, perivascular hemorrhage, inflammatory cell infiltrations and fibrosis, with widening of interstitial spaces of cardiomyocytes, loss of muscle cells continuity and some hyaline degeneration. At the molecular levels, the expression levels of miRNA 328, miRNA-21, TGFβ1, TGFβ1R, JAK1, STST-3, SMAD-1 and NFκβ genes were significantly up-regulated, whereas the expression levels of SMAD-7 were significantly down-regulated. It is concluded that SeNPs and/or BV administration ameliorates the deleterious effects resulting from STZ administration through improving the biochemical, histopathological and molecular effects, suggesting their protective role against the long-term diabetic complications of DCM and DN.

## 1. Introduction

Diabetes mellitus (DM) is a chronic endocrine metabolic disorder of the pancreas, affecting the metabolism and characterized primarily by persistent hyperglycemia, polyuria, polydipsia and polyphagia [1]. DM results mainly from insufficient insulin secretion or insulin resistance and exists in two main types: insulin-dependent diabetes mellitus (IDDM) (Type I) and non-insulin-dependent diabetes mellitus (NIDDM) (Type II) [2].

Diabetic complications, which are mainly due to persistent hyperglycemia [3], are the main life-threatening factor for diabetic patients and affect several organs such as neurons, eyes, heart and kidneys; if not dealt with in time, they may lead to diabetes-associated cardiomyopathy, nephropathy, neuropathy and retinopathy [4,5].

Persistent hyperglycemia was reported to induce diabetic cardiomyopathy (DCM) through a disturbance in oxidant/antioxidant status that resulted in free-radical formation which affects cardiac muscles, resulting in hypertrophy, stiffness and ventricular fibrosis that ends with cardiomyopathy and heart failure [6,7]. Changes in the plasma levels of cardiac biomarkers reflect several myocardial metabolic changes [8]. One third of diabetic patients were reported to develop diabetic nephropathy (DN), which is considered the leading cause of renal failure [9,10,11]. DN is characterized by structural and functional abnormalities [12,13] including albuminuria [14], basement membrane thickening and accumulation of extracellular matrix, and hypertrophy with loss of glomerular epithelial cells [13]. Measurement of plasma urea and creatinine is widely regarded as a test of renal function [15].

C-reactive protein (CRP) is an acute phase protein considered to be an important inflammatory biomarker in the first stage of inflammation in diabetic patients [16,17]. It is induced in response to tissue damage and inflammation [18]. It can induce IL-6 through an NF-κB-dependent mechanism [19].

Several molecular proteins and signaling pathways have been implicated as an important marker for the development of DCM and DN, including Micro RNAs (miRNA), Janus kinase (JAK) signal transducer and activator of transcription factor proteins (STATs) (JAK/STAT), transforming growth factor β1 (TGF-β1)/SMAD and TGF-β/NF-kβ pathways [20,21,22,23,24,25].

miRNAs are short (17–22) nucleotide non-coding RNA molecules, binding to 3’ untranslated regions (UTR) of many target genes and regulating their expression [26,27]. They are involved in the regulation of many cellular processes including differentiation, proliferation and apoptosis [28,29]. The potential role of miRNAs in various diseases has been recently confirmed as a new diagnostic and therapeutic target for use as a potential alternative for disease treatment [30]. Several miRNAs, such as miR-21 and miR-328, were reported to exhibit abnormal expression in DCM and DN [20,31,32]. miR-21 is one of the most studied miRNAs whose expression is significantly changed in DN and DCM [33]. The role of miR-21 in DN and DCM has been studied by many researchers [34,35,36,37]. It is not surprising that miR-21 up-regulation leads to mitochondrial dysfunction and oxidative stress [38,39], with a positive correlation between miR-21 level and CRP concentrations that suggests the inflammatory condition accompanying the diabetic patient [40]. Another miRNA also found to be up-regulated during cardiac hypertrophy is miR-328 [41]. It was found that miR-328 is the most expressed miRNA in atrial fibrillation (AF) patients and is considered a critical point for the transformation of the heart from a state of controlled hypertrophy to a state of uncontrolled cardiac hypertrophy [42,43]. Thus, our main purpose in this study was based on the fact that targeting miRs may help in understanding new prognostic and therapeutic targets in DCM and DN.

TGFβ1, JNKs, STATs, NFκβ and SMADs are the main important targeted genes controlling cardiac growth, remodeling and hypertrophy, as well as the expression of pro-fibrotic genes in cardiac and renal tissues [44,45]. Therefore, they were the main targets in our study.

Numerous anti-diabetic drugs have been used for diabetic treatment; however, there are no drugs without side effects or economic burdens to the patient. Therefore, research is directed toward natural remedies as well as the new trend of nanoparticle use, especially if these products can help to minimize the long-term complications of diabetes.

Previous reports suggested the efficacy of SeNPs and BV as hypoglycemic tools in diabetic rats [4,46,47,48]. SeNPs were found to produce an effective improvement in beta cells and have the ability to restore damaged pancreatic cells [4]. Se itself was reported to have insulin-like action in the cell due to its ability to translocate the glucose receptors into the cell membrane; however, due to its being needed specifically in trace amounts, attention should be paid to its toxicities, so the use of the nanoparticle form may be a way out of this predicament [46]. BV is a complex mixture of peptides and proteins secreted by bees [47,48]. Its main active product, melittin, exhibited several important cellular effects such as cytotoxicity, membrane depolarization, and hemolysis with activation of phospholipases A2 and C. It is also reported to suppress many signaling pathways, such as the NFκβ pathway, thus leading to reduction of inflammation in many tissues [49,50]. BV has been suggested for use as a drug for its therapeutic advantages for neurological, hematological and cardiovascular diseases [47]. It is believed that BV has hypoglycemic and hypolipidemic effects and is able to counteract antioxidant effects [48].

The novelty of our research is to examine the expected role of SeNPs and BV in ameliorating the deleterious effects that result as long-term complications of diabetes through evaluating their biochemical effects on serum cardiac and renal markers, and histopathological effects on cardiac and kidney tissues; we also undertook molecular evaluation of their effects on miR-21 and miR-328, the miRNAs closely related to cardiac and renal fibrosis, with a molecular evaluation of their regulated genes (TGFβ1, JNKs, STATs, NFκβ and SMADs) in cardiac and renal tissues. These findings may be useful to drive insights into the use of nanoparticles and natural products rather than chemically synthetic drugs.

## 2. Materials and Methods

### 2.1. Materials

SeNPs (80 nm, 0.15 wt.% in water dispersion), STZ, DEMSO (anhydrous, ≥99.9%), chloroform, isopropanol and 70% ethanol (HPLC grade) were purchased from Sigma Chemicals Co. (St. Louis, MO, USA)

Bee venom (*Apis mellifera carnica*) specimens were obtained from the Plant Protection Research Institute, Beekeeping Research Department, Agriculture Research Centre in Dokki, Giza, Egypt.

### 2.2. Experimental Animal Housing, Treatment and Grouping

Fifty male adult albino rats weighing (110 ± 20 gm.) at the beginning of the experiment were purchased from the laboratory animal house, Faculty of Vet. Med., Zagazig University, Egypt. Animals were housed in groups of five per cage (two cages for each experimental group) with a relative humidity (55 ± 10%) and 12 h light/dark cycle, with free access to a standard diet and water. Animals from all groups were kept under similar environmental conditions and with the same diet with free access to food and water all through the experimental period. Animal care and cleaning was completed for all animals twice daily. The rats were subjected to ten days of a standard acclimatization period.

### 2.3. Animal Grouping

Animals were randomly divided into five groups (10 rats each). Group (I), control non-diabetic group, did not receive any type of treatment until the start of treatment in other groups, when it received oral gavage of 0.5 mL of 1% aqueous DMSO daily for 4 weeks. The other 40 rats were treated with STZ for induction of diabetes, as will be mentioned below, then divided into 4 different groups. Group (II), diabetic control group: after the start of treatment in other groups, these rats received only oral gavage of 0.5 mL of 1% aqueous DMSO daily for 4 weeks. Group (III), diabetic treated with SeNPs: animals received oral gavage of SeNPs (0.1 mg/kg) in 0.5 mL of 1% aqueous DMSO daily for 4 weeks, following Al-Quraishy et al. [51]. Group (IV), diabetic treated with bee venom (BV): animals received I/P injections of BV (0.05 mg/kg) in PBS 3 times per week for 4 weeks [52]. Group (E), diabetic treated with SeNPs and BV: (N = 10) animals received oral gavage of SeNPs (0.1 mg/kg) daily and BV (0.05 mg/kg) 3 times per week for 4 weeks.

### 2.4. Induction and Assessment of Diabetes

A single I/P injection of 55 mg/kg STZ, prepared in a citrate buffer (pH 4.4, 0.1 M), was used for induction of diabetes, following Bhatt and Addepalli [53]. The control rats received an equal volume of citrate buffer. Blood samples from the retro-orbital plexus were collected 48 h after injection to ensure the incidence of diabetes. The rats were considered to be diabetic if their serum glucose levels were more than 250 mg/dL. The diabetic rats were kept for 5 weeks after induction of diabetes to ensure DCM and DN onset [53]. Then, the treatment with SeNPs and BV started according to the above-mentioned doses.

### 2.5. Blood Sampling and Tissue Collection

Blood samples were withdrawn with a retro-orbital puncture at the end of the experimental period, collected and used for separation of serum by centrifugation at 3000 rpm for 15 min for biochemical determination of serum glucose and insulin concentrations, kidney biomarkers (serum BUN, creatinine, albumin), CRP concentrations and cardiac biomarkers (AST, CK-MB, LDH, Troponin I (cTnI) and Troponin T (cTnT) concentration).

The kidneys and hearts of sacrificed animals were rapidly removed and stored at −80 °C for molecular and RT-PCR analyses (*n* = 3). Additionally, parts from kidneys and hearts of three rats (*n* = 3) from each group were washed in normal saline, isolated in 10% formalin and fixed for histopathological evaluation.

### 2.6. Evaluation of Biochemical Parameters

Serum glucose concentration was determined using Vitro Scient kits following Tietz [54]; serum insulin concentration was determined with IMMULITE kits following Chevenne et al. [55], serum BUN concentration was determined using the Urease-Berthelot Method following Fawcett and Soctt [56], serum creatinine concentration using the colorimetric kinetic method following Bartles et al. [57], serum albumin using the colorimetric method following Doumas et al. [58], serum CRP using a rat CRP-ELISA kit (Catalog No. 41-CRPRT-E01, (Invitrogen; Thermo Fisher Scientific, Inc., Waltham, MA, USA)) following Eckersall [59], serum AST using a colorimetric method kit of Bio-diagnostic Co. (29 El-Tahrer St., Dokki, Giza, Egypt) With Catalog Number AT 10 34 (45) following Reitman and Frankel [60], serum creatine kinase-MB (CK-MB) using an ab285275—Rat Creatine Kinase MB ELISA Kit (Abcam, 24 rue Louis Blanc, Paris, France), serum lactate dehydrogenase (LDH) using an ab102526 Lactate Dehydrogenase (LDH) Assay Kit (Colorimetric), serum cTnI using an ab246529 Rat Cardiac Troponin I Simple Step ELISA^®^ Kit, and serum cTnT using a Rat Cardiac Troponin T (cTn-T) ELISA Kit (Catalog No. CSB-E16443r, (Invitrogen; Thermo Fisher Scientific, Inc., Waltham, MA, USA).

### 2.7. Evaluation of Micro RNAs (miR-328a and miR21) and Other Gene Transcriptional Levels in Renal and Cardiac Tissues

miRNA was extracted from renal and cardiac tissues using a miRNeasy Mini kit (50) from Qiagen with Catalog No. 217004 (Qiagen, Germantown, MD, USA), following manufacturer instructions for purification of microRNA from tissues and cells. Total RNA was extracted from the tissues using Trizol (Invitrogen; Thermo Fisher Scientific, Inc., Waltham, MA, USA), following manufacturer instructions.

The purity of both extracted miRNA and total RNA samples was checked using a spectrophotometer from NanoDrop technologies, (Wilmington, DE, USA). Only samples with a purity of 1.8 or more were used for RT-PCR analysis.

A total of 50 ng of the total extracted RNA was reverse-transcripted in a final volume of 20 µL (50 ng dissolved in 5 µL nuclease-free water, 4 µL of 5× miRCURY RT reaction buffer, 2.5 µL of 10× miRCURY RT Enzyme Mix, 1.2 µL of a primer table, and 10 µL of RNase-free water) in a thermal cycler under the conditions of 42 °C for 60 min, followed by 95 °C for 5 min for enzyme inactivation, according to manufacturer instructions.

The real-time (RT-PCR) was performed in a Rotor-Gene Q Real-Time PCR System (Qiagen, Hilden, Germany) using SYBR Green with low ROX kit with a Cat. No. P725 or P750 (Enzynomics, Daejeon, Republic of Korea).

Forward and reverse primer sequences for miRNA genes and the detected genes in this experiment were synthesized by Sangon Biotech (Beijing, China) and described in Table 1. The conditions for PCR included an initial denaturation at 95 °C for 12 min, followed by 40 cycles of denaturation at 95 °C for 20 s, annealing at 60 °C for 30 s, and extension at 72 °C for 30 s. A melting curve analysis was performed following PCR amplification. The detection method used for calculating the relative quantitation of the transcripts of the replicates with regard to the housekeeping genes was used as a constitutive control for normalization [61].

### 2.8. Histopathological Studies

Specimens of kidney and heart muscles were collected and immediately fixed in 10% neutral buffered formalin for 48 h. The specimens were processed histologically, dehydrated in ascending grades of ethanol, cleared in xylene and embedded in paraffin wax forming paraffin blocks. Five-micron-thick sections were obtained and stained with Harris’s Hematoxylin and Eosin (H&E) following Suvarna et al. [62]. The microphotographs were taken using a digital Dsc-W 130 super steady cyper shot camera (Sony, Tokyo, Japan) connected to an Olympus BX 21 light microscope (Olympus, Tokyo, Japan).

### 2.9. Statistical Analysis

The obtained data were analyzed and graphically represented using the Statistical Package For Social Science (SPSS, 18.0 software, 2011, IBM, Armonk, NY, USA) for obtaining means and standard error. The data were analyzed using one-way ANOVA to determine the statistical significance of differences among groups. Duncan’s test was used for making multiple comparisons among the groups for testing the inter-grouping homogeneity.

## 3. Results

### 3.1. The Effects of SeNPs and/or BV on Biochemical Parameters in the Serum of STZ-Diabetic Rats

STZ administration resulted in a significant hyperglycemia associated with a significant hypo-insulinemia suggesting β-cell destruction. Serum BUN, serum creatinine and CRP concentration were also significantly increased with a significant decrease in the serum in albumin concentration, which suggests the incidence of renal disease due to STZ administration. STZ administration also resulted in a significant increase in the serum CK-MB, AST, LDH, cTnI and cTnT, indicating the incidence of cardiac dysfunction. SeNP and/or BV treatments improved the deleterious effects induced with STZ administration by improving the serum insulin concentrations, which resulted in decreasing the hyperglycemia, decreasing the serum BUN, creatinine and CRP while increasing the serum albumin concentrations. At the same time, SeNP and/or BV administration resulted in a significant decrease in the serum CK-MB, AST, LDH, cTnI and cTnT concentration, suggesting the improvement abilities of these compounds for cardiac and renal function parameters.

### 3.2. The Effect of SeNPs and/or BV on the Expression Levels of miR-21 Gene in Renal Tissues, and miR-21 and miR-328 Genes in Cardiac Tissues

As shown in Figure 1, STZ administration resulted in a significant increase in the expression levels of miR-21 in renal and cardiac tissue as well as an increase in the expression levels of miR-328 in cardiac tissues, suggesting their role in controlling the process of fibrosis in renal and cardiac cells.

### 3.3. The Effect of SeNPs and/or BV on the Expression Levels of TGF-β1, NF-κβ and SMAD-7 Genes in Renal Tissues

As shown in Figure 2, STZ administration resulted in a significant increase in the mRNA expression levels of TGF-β1 and NF-κβ genes, with a significant decrease in the expression levels of SMAD-7 gene. These results suggest the involvement of TGF-β1 and NF-κβ genes in the incidence of renal fibrosis and suggest the inhibitory action of SMAD-7 gene to this pathway. Administration of SeNPs and/or bee venom resulted in a significant decrease in the expression levels of TGF-β1 and NF-κβ genes, with a significant increase in the expression levels of SMAD-7 gene. Co-administration of SeNPs and bee venom resulted in the best effect, nearest to the control group.

### 3.4. The Effect of SeNPs and/or Bee Venom on the Expression Levels of TGF-β1, TGF-βR, JAK-1, STAT-3 and SMAD-1 Genes in Cardiac Tissues

As shown in Figure 3, administration of STZ resulted in a significant increase in the mRNA expression levels of TGF-β1, TGF-βR, JAK-1, STAT-3 and SMAD-1 genes in cardiac tissues. These results suggest the induction of cardiomyopathy and cardiac fibrosis as long-term diabetic complications due to STZ administration. Administration of SeNPs and/or bee venom resulted in a significant decrease in the expression levels of these genes in cardiac tissues, suggesting an ameliorative role of SeNPs and bee venom in decreasing the long-term complications resulting from STZ administration in rats. Co-administration of SeNPs and bee venom resulted in the greatest effect, nearest to the control group.

### 3.5. Histopathological Finding of Renal and Cardiac Tissues due to SeNP and/or BV Administration in STZ-Diabetic Rats

Sacrificed rats of Group 1 (control group) clarified normal, intact renal parenchyma without any pathological changes. The renal parenchyma was organized into two main parts: the outer cortex that houses numerous renal corpuscles surrounded with groups of proximal and distal convoluted tubules; and inner medulla housing numerous medullary rays, bundles of straight tubules, collecting ducts and loops of Henle (Figure 4a). Regarding the heart muscle, this group revealed normal and intact cardiomyocyes with normal orientations and directions, without any pathological abnormalities (Figure 4b).

Sacrificed rats of Group (2) suffering diabetes showed serious microscopical changes in both kidney and heart muscle. These changes were most obvious in the renal tissue of both cortex and medulla, which presented with glomerulonephritis of most glomeruli in the renal cortex (Figure 5a), and hypertrophy of some renal glomeruli (Figure 5b), while the renal medulla of some rats’ kidneys revealed perivascular hemorrhage and endotheliosis (Figure 5c). In addition, severe blood vessel dilatations and congestion with focal fibrosis and degeneration of some renal tubules in the renal cortex were observed (Figure 5d). A few cases revealed focal areas of inflammatory cell infiltrations and fibrosis in both renal cortex and medulla (Figure 5e). Other rats exhibited focal vacuolation of mainly fatty change in renal epithelium of some renal tubules (Figure 5f). Focal areas of cystic dilation of some renal tubules with intra-tubular hemorrhage were observed (Figure 5g), with intra- and inter-tubular extravasated erythrocytes (Figure 5h). Hearts of diabetic rats were affected and showed abnormal architecture of their cardiomyocytes, demonstrating widening of the interstitial spaces with discontinuity of some cardiac muscle fibers (Figure 5i), local inflammatory cell infiltrations, vacuolations of variable shape and size in some cardiac myocytes (Figure 5j), and focal areas of cardiac myocyte degenerations (Figure 5k). Some rats revealed cardiac muscles fibers with hyaline degeneration with or without perivascular fibrosis and congestion (Figure 5l).

Sacrificed rats of Group (3) suffering diabetes and treated with SeNPs showed noticeable reduction in previously demonstrated lesions in both organs, while renal lesions were restricted mainly to the renal medulla, which exhibited cystic dilation of some renal tubules with or without hyaline casts inside (Figure 6a,b). Focal infiltration of inflammatory cells was also noticeable (Figure 6c,d). Heart tissues appeared with local infiltration of inflammatory cells (Figure 6e) and with proliferation of Anitschkow cells (Figure 6e). In some examined sections, moderate-to-severe congestion and dilatation were declared in blood vessels (Figure 6f).

Sacrificed rats of Group (4) suffering diabetes and treated with BV showed slight reduction in lesions observed in diseased rats and exhibited renal glomerulonephritis (Figure 7a); hypercellularity of some renal glomeruli of the renal cortex (Figure 7b) with diffuse congestion of blood vessels with cystic dilation and cloudy swelling of some renal tubules with or without atrophy of others within the renal medulla (Figure 7c–e). In addition, diffuse inter- and intra-tubular hemorrhage and congestion of blood vessels with cystic dilation of some renal tubules were demonstrated (Figure 7f). Cardiac muscle revealed degeneration of some fibers with proliferation of Anitschkow cells (Figure 7g), while others showed vacuolation of the vascular wall (Figure 7h).

Sacrificed rats of Group (5) suffering diabetes and treated with both SeNPs and BV had the best improvement in the tissue architecture among the diabetic groups for both kidneys and hearts: the renal cortex exhibited apparently normal glomeruli and renal tubules (Figure 8a). Only a few renal infiltrations of casts were seen in some renal tubules of the renal medulla (Figure 8b). Furthermore, great improvements of cardiac muscle architecture were observed, where the cardiomyocytes were apparently normal and intact with normal orientations (Figure 8c).

## 4. Discussion

The motivation of doing this work arose from the urgent need to minimize the effects of life-threatening long-term complications of diabetes, including DCM and DN, which are considered to be the two main problems annoying diabetic patients due to their fatal effect on hearts and kidneys. Dealing with this problem in time and discovering a new trend diminishing these complications can help us to protect diabetic patients and save their lives.

Several reports have suggested the hypoglycemic effect of SeNPs and BV in diabetic rats, and since the hyperglycemia was the main cause of long-term complications, these products can represent a safe way to deal with long-term diabetic complications and reduce their deleterious effects [4,46,47,48]. This idea motivates us to evaluate the role of SeNPs and/or BV as hypoglycemic agents in the reduction of cardiac and renal fibrosis produced as a long-term diabetic complication, through examining their effects on the expression levels of miR-21 and miR-328, the genes involved in the regulation of cardiac and renal fibrosis, as well as by examining their effects on the genes of JAK/STAT, TGF-β/SMAD and TGF-β/NF-kβ signaling pathways, the main pathways involved in cardiac and renal fibrosis.

### 4.1. Effect of SeNPs and/or Bee Venom on Blood Glucose and Insulin Concentrations

Induction of diabetes in our study was performed using I/P injection of STZ, which was known to induce beta-cell destruction that results in a characteristic change in DM, starting from hyperglycemia with hyperinsulimeia, followed by transient hypoglycemia and finally persistence of chronic hyperglycemia [53,63]. The induced persistent hyperglycemia is the key for oxidative stress production and induction of diabetic complications affecting the nervous system, kidneys and cardiac muscles [64,65]. These suggestions were confirmed by our results, presented in Table 2, which explained the hyperglycemic status with a reduction in insulin levels due to STZ injection.

SeNP and/or BV administration resulted in a reduction of the hyperglycemia produced by STZ administration through an improvement of insulin production. SeNPs were reported to induce this hypoglycemic effect through their beta-cell restoring properties as it helps in regeneration of the pancreatic beta cells and helps in insulin production [53,66]. Zahran et al. also confirmed the hypoglycemic effect of BV against STZ-induced hyperglycemia [67]. The co-administration of SeNPs and bee venom presented the greatest hypoglycemic effect in our work.

### 4.2. Biochemical and Histopathological Effects of SeNPs and/or BV on Renal and Cardiac Functions

Biochemically, the renal and cardiac functions were evaluated through examining some blood parameters such as serum BUN, creatinine and serum albumin for renal function; serum CK-MB, LDH1, AST, cTnI and cTnT for heart function; and CRP levels as an acute phase protein present in an inflammatory status.

STZ injection resulted in a significant increase in the serum levels of BUN and creatinine with a significant reduction in the serum albumin concentration (Table 2). These findings reveled the relationship between hyperglycemia and an increase in BUN and creatinine concentrations and confirm that hyperglycemia is a major cause of progressive renal damage [68]. BUN and creatinine are the simplest tests for evaluation of kidney diseases either in acute or chronic status; they are natural waste products for amino acids and creatine metabolism, and usually their levels are elevated during kidney diseases [15]. Reduction of serum albumin concentration is known to be one of the characteristic markers for both hepatic and renal diseases [69]. In our results, serum albumin concentration accompanied by elevation of serum BUN and creatinine concentrations confirmed the incidence of diabetic kidney diseases induced with STZ.

These results were confirmed by our histopathological analysis of renal tissues which revealed glomerulonephritis and hypertrophy of most glomeruli of diabetic rats, and the renal medulla revealed perivascular hemorrhage, endotheliosis with severe dilatation of blood vessels, congestion, focal fibrosis and degeneration of renal tubules; the complete pathological lesions are illustrated in Figure 5.

As inflammation is considered the key modulator for diabetic pathogenesis and its complications in hearts and kidneys, several studies connected the elevation in CRP levels with the incidence of Type I diabetes [70]. The elevation of CRP levels observed in our experiment may have resulted in response to persistent hyperglycemia occurring in diabetic group, as suggested by many pervious reports [71,72].

The serum cardiac biomarkers (CK-MB, AST, LDH, cTnI, cTnT) were also elevated by STZ injection, which indicates a cardiac dysfunction in the diabetic group. CK-MB and troponins are cardiac biomarkers usually used in diagnosis of cardiac diseases [73]. The elevated CK-MB activity in the diabetic group may be due to disturbances of the respiratory chain, Krebs cycle and other metabolic defects affecting the mitochondria of cardiac cells that result in ATP depletion and decrease in creatine phosphate formation, which may finally result in AMP-activated protein kinase activation as a late complication [74,75]. Cardiac troponins are also considered to be international biomarkers for detection of myocardial diseases, and their serum elevation in the diabetic group confirmed the incidence of myocardial injury [76]. The serum elevation of LDH-1, which are present in four hearts (H) subunits in our results, can be considered biomarkers for cardiac cell damage. Elevation of all these cardiac biomarkers in our experiment, together with the elevation of serum AST, the transaminase of cardiac and hepatic cells, can be considered confirmatory evidence for induction of diabetic cardiomyopathy due to STZ injection.

The cardiac tissue defects of diabetic rats were also confirmed by histopathological investigation that showed abnormal cardiomyocyte architecture represented by widening of the interstitial spaces, discontinuous cardiac muscle fibers, inflammatory cell infiltrations, vacuolations of some cardiac myocytes, and cardiac myocyte degenerations with hyaline degeneration of cardiac muscles fibers with or without perivascular fibrosis and congestion.

According to our knowledge, this is the first work to evaluate the histopathological improvement due to SeNP and/or BV administration to STZ-damaged cardiac and renal tissues. SeNP and/or BV administration resulted in a significant improvement of the serum cardiac and renal biochemical markers as well as histopathological improvement. The results illustrated in Table 2 and Figure 6, Figure 7 and Figure 8 showed that SeNPs and BV reduce the serum BUN, creatinine and all serum cardiac markers with an improvement of the serum albumin levels, whereas the co-administration of both SeNPs and BV resulted in the greatest improvement effects. This improvement in biochemical parameters was reflected in and appeared more clear on histopathological examination, where the renal cortex exhibited apparently normal glomeruli; renal tubules showed only a few renal infiltrations of casts in some renal tubules of the renal medulla; and the cardiomyocytes showed apparently normal, intact cells with normal orientations and normal cardiac muscle architecture. Serum CRP levels were also improved, which reflected the anti-inflammatory effects of SeNPs and BV.

### 4.3. Molecular Role of SeNPs and/or BV

#### 4.3.1. Effect of SeNPS and/or BV on Expression Levels of miR-21 and miR-328 in Renal and Cardiac Tissues

The expression of miR-328 was evaluated in cardiac tissues, whereas the expression of miR-21 was evaluated in both cardiac and renal tissues. Our attention was directed to miR-21 and miR-328 due to several reports connecting them with the pathogenesis of DCM and DN [33,34,35,42,43,44]. STZ injection showed a significant up-regulation in the expression levels of both miRNAs, whereas SeNPs and/or BV succeeded in ameliorating this up-regulation through decreasing the expression levels of miR-328 and miR-21 in both cardiac and renal tissues. It is the first time that effects of SeNPs and/or BV on miR-328 and miR-21 in both cardiac and renal tissues were evaluated. miR-21 is known to induce fibrosis in many organs, including hearts and kidneys [35,36,37], and to promote TGF-β effects on endothelial dysfunction [39]. It was reported to exhibit its pathogenic effect in kidneys by targeting the genes involved in several signaling pathways, such as the TGF-β/SMAD pathway, TGF-β/NF-kβ signaling pathways and others, resulting in epithelial mesenchymal transition (EMT), extracellular matrix deposition (EMD), inflammation and fibrosis [34,45]. Thus, miR-21 is considered to be involved in function impairment, remodeling and fibrosis. Bang et al. also reported the contribution of miR-21 to the development of fibroblast-derived cardiomyocyte hypertrophy [77]. miR-328 is a strong pro-fibrotic miRNA in the heart, governing atrial fibrillation through targeting the genes encoding Ca^2+^ channels [42] and promoting cardiac hypertrophy that is considered the primary predictor of chronic heart diseases and cardiomyopathy [43]. Du et al. [78] demonstrated that miR-328 is a potent pro-fibrotic miRNA involved in cardiac fibrosis through targeting TGFβR as a main pathway regulating cardiac fibrosis, providing a new idea that inhibition of miR-328 may interfere with progression of cardiac fibrosis, and thus promising a new therapeutic strategy for dealing with cardiac fibrosis.

#### 4.3.2. Effect of SeNPs and /or BV on the Expression of Genes Involved in JAK/STAT, TGF-β/SMAD and TGF-β/NF-kβ Signaling Pathways in Renal and Cardiac Tissues

As miR-21 directly controls the genes involved in the TGF-β/SMAD pathway and TGF-β/NF-kβ signaling pathways and consequentially controls fibrous formation in both renal and cardiac tissues [79,80], we investigated the expression levels of the main genes involved in these pathways. There is much evidence that TGF-β signaling pathways play a critical role in the development of diabetic complications [81]. TGF-β is considered a pro-sclerotic cytokine involved in organ fibrosis; its signaling pathway is the primary inducer of EMT in various fibrotic and cancer diseases [82]. TGF-β1 is the principal isoform in the heart, involved with stimulation of cardiac fibrosis; elevated expression of TGF-β1 is observed during transition from a stable hypertrophic state to the heart failure state [44]. JAK/STAT3 signaling is required for TGF-β-mediated transcriptional responses [45]. Previous findings indicated that activation of JAK/STAT signaling is crucial for development of myocardial fibrosis due to its role in controlling diverse processes, including differentiation, proliferation, cellular immunity, inflammation and apoptosis. An activated JAK/STAT signal pathway up-regulated the expression of TGF-β, Type I and Type II collagens, leading to production of fibrosis [83].

There is a little information on whether the JAK/STAT pathway is involved in the protection of SeNPs and/or BV against myocardial fibrosis in diabetes mellitus. In the present work, SeNPs and/or bee venom were used to evaluate their possible anti-fibrotic role in hearts and kidneys of STZ-induced diabetic rats.

In our study, the expression levels of cardiac TGFβ1, TGFβ1R, JAK-1, STAT-3 and SMAD-1 were significantly up-regulated in rats treated with STZ, compared to control rats. In the same way, renal TGFβ-1 and NF-κβ were also significantly up-regulated in the diabetic group, whereas SMAD-7 was significantly down-regulated in the STZ-diabetic group, suggesting its anti-fibrotic role against kidney injury induced by diabetic complications. It is suggested that hyperglycemia and its glycated products may be the main cause that may induce oxidative stress, which results in induction of inflammatory cytokines that follow JAK/STAT and TGF-β signaling activation [84].

The study by Zhong et al. [85] confirmed that SMAD-7 was the principal target of miR-21 during kidney inflammation, and overexpression of miR-21 in the renal tissues directly reduced the expression SMAD-; thus, targeting SMAD-7 may be the mechanism by which miR-21 produces its effects in renal tissues. This explains the down-regulation observed in the expression levels of SMAD-7 due to STZ administration in our experiment, confirming at the same time the inhibitory effect of SMAD-7 toward TGFβ signaling pathway and suggesting a new targeted way for decreasing this diabetic complication.

SeNPs and/or BV succeed in improving the kidney and heart status as observed in histopathological examinations, suggesting that they produce these effects through targeting SMAD-7, as they resulted in a significant increase in the expression levels of the SMAD-7 gene in renal tissues. It is also suggested that the ameliorative effects produced by SeNPs and/or bee venom may be a result of their inhibitory effects affecting the TGF-β1 and its receptor gene expression and the down-regulation happening to JAK, STAT-3, SMAD-1 and NFκβ in cardiac or renal tissues.

## 5. Conclusions

In conclusion, our finding showed that cardiac and renal fibrosis, two complications involved in diabetes, can be attenuated by SeNP and/or BV administration through several biochemical and molecular pathways involving down-regulation of several genes involved in TGF-β1 signaling pathways as well as up-regulation of SMAD-7 expression, which has an inhibitory effect for renal fibrosis. They also showed hypoglycemic and hyper-insulinemic effects, as well exhibiting a major role in improving both renal and cardiac functions through the ability to improve the serum levels of both renal and cardiac biomarkers.

## Figures and Tables

**Figure 1 metabolites-13-00400-f001:**
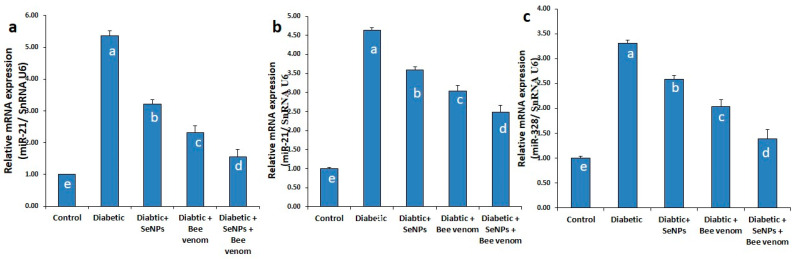
The expression levels of miR-21 in renal tissues (**a**), miR-21 and miR-328 in cardiac tissues (**b**,**c**) in normal and STZ-diabetic rats.

**Figure 2 metabolites-13-00400-f002:**
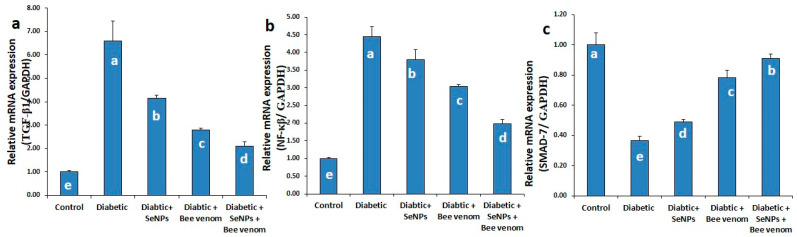
The expression levels of TGF-β1 (**a**), NF-κβ (**b**) and SMAD-7 (**c**) genes in renal tissues of normal and STZ-diabetic rats.

**Figure 3 metabolites-13-00400-f003:**
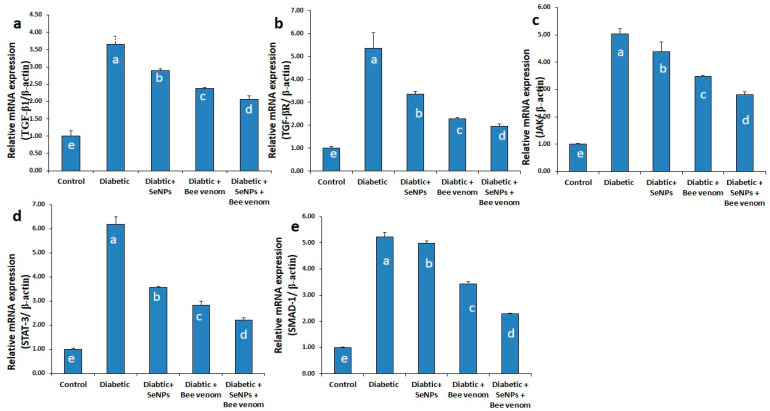
The expression levels of TGF-β1 (**a**), TGF-βR (**b**), JAK-1 (**c**), STAT-3 (**d**) and SMAD-1 (**e**) genes in cardiac tissues of normal and STZ-diabetic rats.

**Figure 4 metabolites-13-00400-f004:**
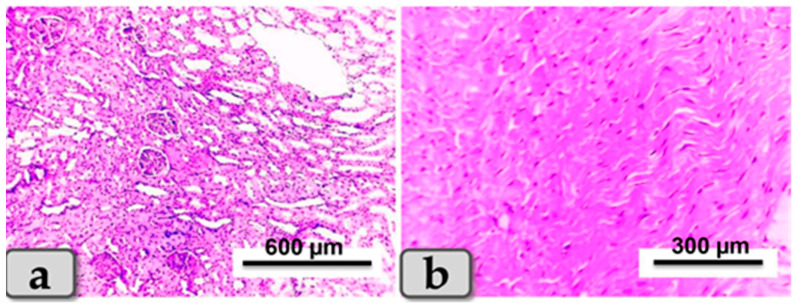
Photomicrograph of a rat’s kidney and heart from the control group (Group 1). (**a**) Showing kidney with normal and intact renal parenchyma, renal cortex and medulla. (**b**) Showing cardiomyocytes with normal, intact cardiac muscle fibers with normal orientations and directions without any pathological lesions. Stain: All H&E.

**Figure 5 metabolites-13-00400-f005:**
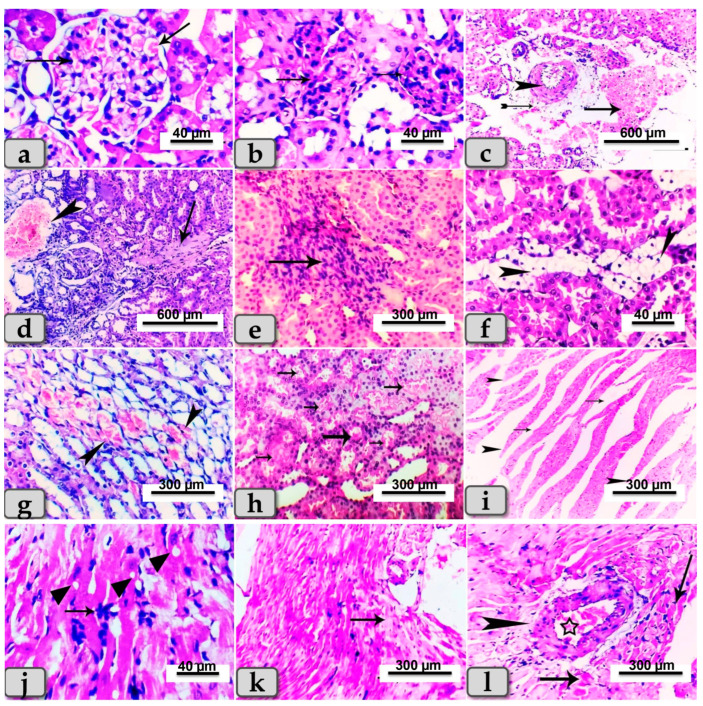
Photomicrographs of kidney and hearts from rats of STZ-diabetic group (Group 2). (**a**) Showing glomerulonephritis of renal cortex (arrows). (**b**) Showing hypertrophy of the renal glomeruli of renal cortex (arrow). (**c**) Showing perivascular hemorrhage of renal medulla (arrow), fibrosis (tailed arrow) and endotheliosis (arrowhead). (**d**) Showing severe blood vessel dilatations and congestion (arrow head) with focal fibrosis (arrow) and degeneration of some renal tubules in the renal cortex. (**e**) Showing focal area of inflammatory cell infiltrations and fibrosis in the renal medulla (arrow). (**f**) Showing focal vacuolations (mainly fatty change) of the renal lining epithelium of some renal tubules (arrowhead). (**g**) Showing focal area of cystic dilation with intra-tubular hemorrhage (arrow head). (**h**) Showing intra-tubular and inter-tubular extravasated erythrocytes (arrows). (**i**) Photomicrograph of cardiomyocytes with widening of interstitial spaces (arrows) and discontinuity of some cardiac muscle fibers (arrowhead). (**j**) Showing inflammatory cell infiltrations (arrows) and vacuolations of variable shape and size in some cardiac myocytes (arrow head). (**k**) Showing focal areas of cardiac myocyte degenerations (arrows). (**l**) Showing hyaline degeneration of some cardiac myocytes (arrows) with perivascular fibrosis (arrow head) and congestion (star). Stain: All H&E.

**Figure 6 metabolites-13-00400-f006:**
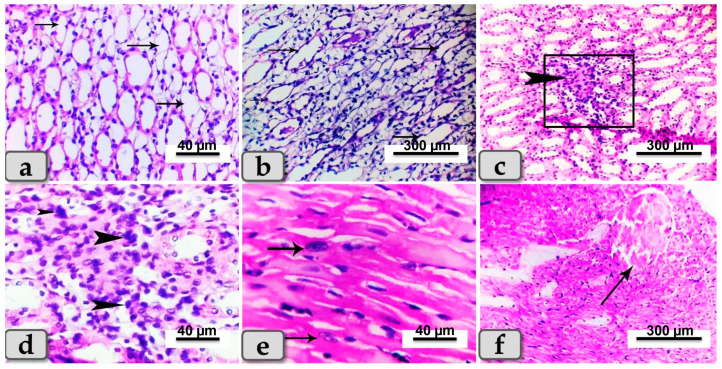
Photomicrographs of kidneys sand hearts from rats suffering diabetes and treated with SeNPs (Group 3). (**a**) Showing cystic dilation of a few renal tubules (arrows). (**b**) Showing cystic dilation of a few renal tubules (arrows) with hyaline casts inside. (**c**) Showing focal infiltration of inflammatory cells in renal medulla (arrow head). (**d**) Higher magnification of the previous figure demonstrating the infiltrated inflammatory cells (arrow head). (**e**) Photomicrograph of the cardiac myocytes showing proliferation of Anitschkow cells (arrows). (**f**) Showing blood vessel dilatation and congestion (arrow). Stain: All H&E.

**Figure 7 metabolites-13-00400-f007:**
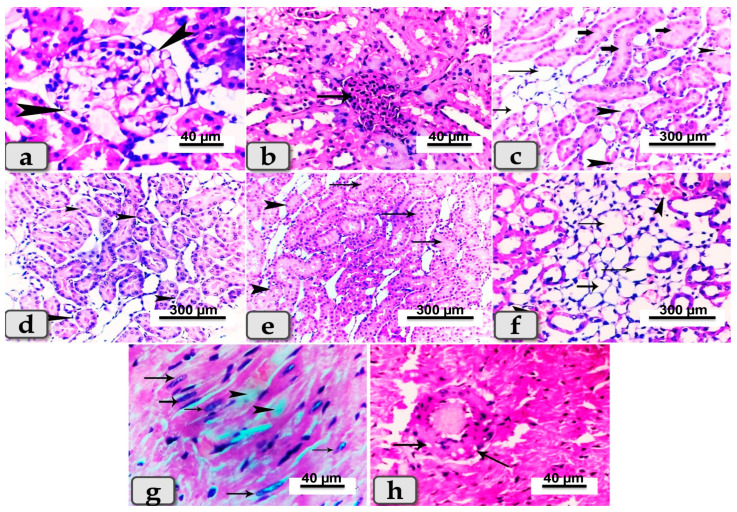
Photomicrographs of kidneys and hearts from rats suffering diabetes and treated with BV (Group 4). (**a**) Showing renal glomerulonephritis (arrow heads). (**b**) Showing hyper-cellularity of renal glomeruli (arrow). (**c**) Showing diffuse congestion of blood vessels (arrow head) with cystic dilation of some renal tubules (thin arrows) and cloudy swelling of others (thick arrow). (**d**) Showing atrophy of some renal tubules within renal medulla (arrow head). (**e**) Showing cloudy swelling of some renal tubules (arrows) and atrophy of others (arrow head). (**f**) Showing diffuse inter- and intra-tubular hemorrhage and congestion of blood vessels (arrow head) with cystic dilation of some renal tubules (arrows). (**g**) Photomicrograph of cardiac muscle showing degeneration of some cardiomyocytes (arrow head) and proliferation of Anitschkow cells (arrows). (**h**) Showing vacuolations of the vascular wall (arrows). Stain: All H&E.

**Figure 8 metabolites-13-00400-f008:**
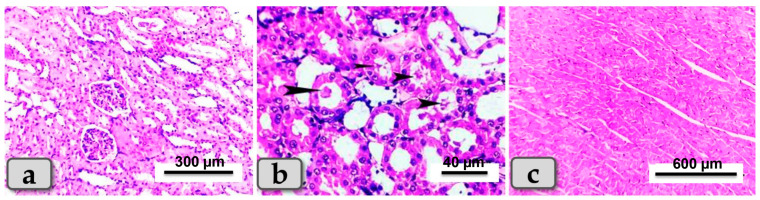
Photomicrographs of kidneys and hearts from rats suffering diabetes and treated with both SeNPs and BV (group 5). (**a**) Showing renal cortex of apparently normal, intact renal tubules and renal glomeruli. (**b**) Showing a few infiltrations of casts in some renal tubules of renal medulla (arrow head) and normal intact tissue. (**c**) Showing apparently normal and intact cardiac myocytes. Stain: All H&E.

**Table 1 metabolites-13-00400-t001:** Forward and reverse primer sequences for miRNA genes and the detected genes.

Gene	Primers	Organ	Accession No.	Expected Size
TGFβ1	5′-AGGGCTACCATGCCAACTTC-3′ 5′-CCACGTAGTAGACGATGGGC-3′	Heart and kidney	NM_021578.2	186
TGFβR	5′-CAGGAGGTGAAAGTCCCCG-3′ 5′-CACTGTTCAACTTGCCTCGC-3′	Heart	NM_017256.1	172
JAK-1	5′-AGGAATGTACTGGGCGTCTT-3′ 5′-GGTTGTCCAGTGTCCCTGAAA-3′	Heart	NM_053466.1	107
STAT-3	5′-GGTACAATCCCGCTCGGTG-3′ 5′-AGCTGGTTCCACTGAGCCAT-3′	Heart	NM_012747.2	169
SMAD-1	5′-TCAATAGAGGAGATGTTCAAGCAGT-3′ 5′-GAAACCATCCACCAACACGC-3′	Heart	NM_013130.3	134
β-actin	5′-CCCGCGAGTACAACCTTCTT-3′ 5′-CGCAGCGATATCGTCATCCA-3′	Heart	NM_031144.3	83
NF-*к*B	5′-CAGGACCAGGAACAGTTCGAA-3′ 5′-CCAGGTTCTGGAAGCTATGGAT-3′	Kidney	NM_199267.2	150
SMAD7	5′-GAGTCTCGGAGGAAGAGGCT-3′ 5′-CTGCTCGCATAAGCTGCTGG-3′	Kidney	NM_030858.2	84
GAPDH	5′-GCATCTTCTTGTGCAGTGCC-3′ 5′-GGTAACCAGGCGTCCGATAC-3′	Kidney	NM_017008.4	91
rno-miR-21-5p	RT primer 5′-GTCGTATCCAGTGCAGGGT- CCGAGGTATTCGCACTGGATACGACTCAACA-3′	F5′-AGCGACTAGCTTATCAGACT-3′ R 5′-GTCGTATCCAGTCAGGGT-3′		
rno-miR-328a-5p	RT primer 5′-GTTGGCTCTGGTGCAGGGT- CCGAGGTATTCGCACCAGAGCCAACTGAGCC-3′	F 5′-GTTTTTGGGGGGCAGGAG-3′ R 5′-GTGCAGGGTCCGAGGT-3′		
snRNA U6	RT primer 5′-AACGCTTCACGAATTTGCGT-3′	F 5′-CTCGCTTCGGCAGCACA-3′ R 5′-AACGCTTCACGAATTTCG-T-3′		

**Table 2 metabolites-13-00400-t002:** The effect of SeNPs and/or BV on biochemical parameters related to heart and kidneys of normal and STZ-diabetic rats.

	Control	Diabetic	Diabetic + SeNPs	Diabetic + Bee Venom	Diabetic + SeNPs + Bee Venom
Glucose conc. (mg/dL)	167.51 ± 9.37 ^c^	334.05 ± 13.68 ^a^	261.18 ± 11.20 ^b^	233.27 ± 12.28 ^c^	206.04 ± 11.39 ^d^
Serum insulin conc. (µlU/mL).	1.63 ± 0.24 ^a^	0.45 ± 0.04 ^e^	1.06 ± 0.20 ^c^	0.76 ± 0.12 ^d^	1.41 ± 0.09 ^b^
Serum BUN (mg/dL)	14.25 ± 1.67 ^d^	33.65 ± 3.89 ^a^	23.13 ± 3.2 ^b^	21.88± 2.94 ^b^	17.41 ± 1.64 ^c^
serum creatinine conc. (mg/dL)	1.1 ± 0.02 ^d^	1.89 ± 0.23 ^a^	1.43 ± 0.18 ^b^	1.38 ± 0.08 ^b^	1.21 ± 0.04 ^c^
Serum albumin (gm%)	6.1 ± 0.23 ^d^	2.65 ± 0.12 ^a^	3.74 ± 0.27 ^b^	3.82 ± 0.21 ^b^	4.68 ± 0.22 ^c^
serum CRP (ng/mL)	4.85 ± 0.39 ^d^	11.52 ± 1.42 ^a^	8.42 ± 1.06 ^b^	7.95 ± 1.1 ^b^	5.42 ± 0.68 ^c^
serum CK-MB (IU/L)	184.5 ± 9.45 ^d^	862.8 ± 19.6 ^a^	683.2 ± 18.7 ^b^	582.5 ± 9.1 ^b^	312.8 ± 10.8 ^c^
serum AST conc. (IU/L)	151.44 ± 11.38 ^e^	521.33 ± 19.83 ^a^	411.2 ± 18.6 ^b^	378.54 ± 16.1 ^c^	225.28 ± 13.53 ^d^
serum LDH conc. (IU/L)	320.5 ± 8.5 ^e^	809.8 ± 15.5 ^a^	511.9 ± 26.5 ^b^	491.6 ± 16.5 ^c^	403.4 ± 12.7 ^d^
cTnI (ng/mL)	0.65 ± 0.15 ^e^	1.83 ± 0.18 ^a^	1.29 ± 0.08 ^b^	1.26 ± 0.09 ^c^	0.94 ± 0.07 ^d^
cTnT (pg/mL)	42.81 ±5.42 ^e^	276.8 ± 21.8 ^a^	182.9 ± 9.7 ^b^	167.6 ± 11.6 ^c^	88.4 ± 7.52 ^d^

Means within the same rows carrying different superscripts (^a^, ^b^, ^c^, ^d^, ^e^) are significant at (*p* ≤ 0.05).

## Data Availability

Data is contained within the article.

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
