# Peer review of "Effect of Selenium Nanoparticles and/or Bee Venom against STZ-Induced Diabetic Cardiomyopathy and Nephropathy"

_metabolites, 2023, doi:10.3390/metabo13030400_

Round 1

Reviewer 1 Report

Dear Authors,

I reviewed the research article entitledEffect of Selenium nanoparticles and/or Bee venom against STZ-induced diabetic cardiomyopathy and nephropathy”. This study demonstrates the results of therapy diabetic rats by selenium nanoparticles and Bee venom and has great potential and practical significance.

The study is well-organized, but several issues must be added or corrected.

My recommendations and suggestions to the Authors are listed below:

1. P. 1, Abstract.

I kindly recommend the Authors to define STZ because the term Streptozotocin is not weidely known.

2. P. 3, section 2, subsection 2.1, line 109

The mean diameter and standard deviation (SD) of nanoparticles used must be added.

3. P. 3, section 2, subsection 2.1

DMSO should be listed in the Materials section, as well as its grade.

4. P. 5, line 198, Table 1.

I kindly recommend to analyze also the JAK-2 and JAK-3 apart from JAK-1 in your further investigations.

5. P. 6, line 229, Table 2.

I kindly recommend the Authors to plot the graphs or diagrams based on this table (in addition to the table). The visual demonstration is preferable for understanding.

The indexes a,b,c,d,e should be described at the bottom of the Table.

Moreover, Table 2 should be cited in the main text (subsection 3.1, line 214).

Please, be attentive with the references: the Tables and Figures must be cited as it demonstrated in the MDPI template.

6. P. 7, line 239, Figure 1

Quality of figures should be improved. Please, add the descriptions of A, B, C into the caption.

Please, add the reference to this Figure from the main text.

7. P. 7, line 252, Figure 2

Quality of figures should be improved. Please, add the descriptions of A, B, C into the caption.

Please, correct the reference: Figure should be written from Capital letter.

8. P. 8, line 266, Figure 3

Quality of figures should be improved. Please, add the descriptions of A, B, C into the caption.

Please, correct the reference: Figure should be written from Capital letter.

I kindly recommend the Authors to use the MDPI template.

9. P. 8, line 275

Please, correct the "Fig." into "Figure". Please, correct the same misprint in the whole text.

10. P. 8, line 278, Figure 4

I kindly recommend adding the scale bar value into the microphotograph. There is a lot of free space in the pictures for the value (300 or 40 um). The value written in the caption is not suitable.

Please, check all photomicrographs for similar issues.

11. P. 13, Line 434

It would be illustrative, if the Authors demonstrate the summarized comparative table with information related to the histopathological effects which were caused by STZ injection. I think it would help to improve the visual attractivity of the paper.

12. P.  15, line 522

The individual detailed Conclusion is recommended.

13. Please, fill the sections: Author Contributions, Funding, Institutional Review Board Statement, Informed Consent Statement, Data Availability Statement, Acknowledgments, Conflicts of Interest.

14. P. 15, line 564

Please, use the MDPI style of references. Be attentive with the Bold text and Italic text.

Based on the aforementioned, I suggest minor revisions prior to the acceptance of this publication.

Author Response

Dear professor,

Thank you for your valuable comments, we are glad to respond to the comments in this file 

1st -reviewer comments

Site of comment

comment

Revision

Abstract

The first phrase does not make sense

Removed

Line 102

the present investigation aimed to evaluate

Replaced

Please follow the recommended structure in Instructions for Authors (e.g. name of section 2 is Materials and Methods, not Experimental design)

Corrected

Abstract

This part should contain the main purpose of your study, which, in your case, is not clear. It is important that Abstract follows the structure of the manuscript. In the present form it has too many generalities and lacks the purpose. Please modify accordingly and check the whole abstract for the recommended number of words in the recommendations for authors

Whole abstract is revised, the main aim is added and the number of words is checked.

Introduction

You offer a documented background of your study but you need to compare the purposes of your study with similar ones (which you do not quite present) and clarify what you bring in novelty. It is very important to state what exactly you bring in novelty in order to express your originality. The purpose of the study needs to be found in the last paragraph and be clearer. Please add. In the present form, as neither Abstract, nor Introduction describe the purpose adequately, the manuscript lacks one of its most important parts. Please add further information and justifications and modify accordingly

The introduction is revised, novelty is cleared, the purpose of the study is cleared and fully modified in the last paragraph

Discussions

As you also compare your results with similar ones obtained in literature, you should state once again what your study brings in novelty, this time in terms of results

Discussion is revised and the novelty is added to comparisons

Conclusions

Please add this section, together with perspectives of your study

Added

References

do not follow the recommendations of MDPI journals, please check Guidelines for Authors and modify accordingly

All references were Corrected

Reviewer 2 Report

General comments

-        Overall, in my opinion the topic of this research work is not suitable for “Metabolites”. At least, I would expect to see some analytical results regarding the chemical characterization of the Bee venom, and possible correlations between specific chemical constituents and the observed effects of treatment in animals.

-        Why the SZT-treated mice model was chosen? How can you assure that the results you obtained were not due to interactions between SZT and the administered nanoparticles and venom bee?

-        Writing English should be carefully revised. Several errors were found throughout the manuscript.

Abstract

-        The meaning of STZ should be reported.

-        Lines 20-28: in my opinion it is better to report more in detail the effects of treatment, instead of the deleterious effects of STZ.

Introduction

-        Lines 88-93: recent reports regarding the use of SeNPs in diabetes should also be introduced, such as: doi.org/10.3390/molecules27175642

Methods

-        Lines 122-25: The approval code of the animal trial should be reported.

-        Figure resolution must be improved. Also, the use of letters to indicate the significance of the results needs to be explained in the figure captures.

Author Response

2nd reviewer - Comments

Site of comment

Comment

Revision

P. 1, Abstract

I kindly recommend the Authors to define STZ because the term Streptozotocin is not weidely known

The term is added

P. 3, section 2, subsection 2.1, line 109

The mean diameter and standard deviation (SD) of nanoparticles used must be added

Added

P. 3, section 2, subsection 2.1

DMSO should be listed in the Materials section, as well as its grade

Added

P. 5, line 198, Table 1.

I kindly recommend to analyze also the JAK-2 and JAK-3 apart from JAK-1 in your further investigations

Thank you for your suggestion, it will be considered in the following investigations

P. 6, line 229, Table 2.

I kindly recommend the Authors to plot the graphs or diagrams based on this table (in addition to the table). The visual demonstration is preferable for understanding.

The indexes a,b,c,d,e should be described at the bottom of the Table.

Moreover, Table 2 should be cited in the main text (subsection 3.1, line 214).

Please, be attentive with the references: the Tables and Figures must be cited as it demonstrated in the MDPI template.

Thank you for your recommendation, the table cited in the main text and more clarified, but I think duplication of table and figure for the same result can be too much, and due to several biochemical parameters I preferred to collect it in a table

a,b,c,d,e letters were added

P. 7, line 239, Figure 1

Quality of figures should be improved. Please, add the descriptions of A, B, C into the caption.

Please, add the reference to this Figure from the main text.

Quality of the figure improved and the descriptions of the letters were added, reference added

P. 7, line 252, Figure 2

Quality of figures should be improved. Please, add the descriptions of A, B, C into the caption.

Please, correct the reference: Figure should be written from Capital letter.

Quality of the figure improved and the descriptions of the letters were added, reference added and the capital letter added

P. 8, line 266, Figure 3

Quality of figures should be improved. Please, add the descriptions of A, B, C into the caption.

Please, correct the reference: Figure should be written from Capital letter.

I kindly recommend the Authors to use the MDPI template.

Quality of the figure improved and the descriptions of the letters were added, reference added and the capital letter added

P. 8, line 275

Please, correct the "Fig." into "Figure". Please, correct the same misprint in the whole text.

10. P. 8, line 278, Figure 4

I kindly recommend adding the scale bar value into the microphotograph. There is a lot of free space in the pictures for the value (300 or 40 um). The value written in the caption is not suitable.

Please, check all photomicrographs for similar issues.

The "Fig." is corrected into Figure in whole text.

Scale bar added into microphotograph. And the values removed from caption

. P. 13, Line 434

It would be illustrative, if the Authors demonstrate the summarized comparative table with information related to the histopathological effects which were caused by STZ injection. I think it would help to improve the visual attractivity of the paper.

Thank you for your advice, connection is done

P.  15, line 522

The individual detailed Conclusion is recommended

Conclusion is added

P. 15, line 564

Please, use the MDPI style of references. Be attentive with the Bold text and Italic text.

References is corrected

Reviewer 3 Report

Dear Authors,

The present study evaluates the effect of Selenium nanoparticles and/or Bee venom against STZ-induced diabetic cardiomyopathy and nephropathy. The research subject is interesting and brings scientific important data in the field, as it deals with a subject that is currently of great interest. Some changes of the manuscript should nevertheless be performed in order to improve its quality. Following specific changes should thus be performed:

 Minor changes

Abstract: The first phrase does not make sense.

Line 102: the present investigation aimed to evaluate

 Major changes

Please follow the recommended structure in Instructions for Authors (e.g. name of section 2 is Materials and Methods, not Experimental design).

Abstract: This part should contain the main purpose of your study, which, in your case, is not clear. It is important that Abstract follows the structure of the manuscript. In the present form it has too many generalities and lacks the purpose. Please modify accordingly and check the whole abstract for the recommended number of words in the recommendations for authors.

Introduction: You offer a documented background of your study but you need to compare the purposes of your study with similar ones (which you do not quite present) and clarify what you bring in novelty. It is very important to state what exactly you bring in novelty in order to express your originality. The purpose of the study needs to be found in the last paragraph and be clearer. Please add. In the present form, as neither Abstract, nor Introduction describe the purpose adequately, the manuscript lacks one of its most important parts. Please add further information and justifications and modify accordingly.

Discussions: As you also compare your results with similar ones obtained in literature, you should state once again what your study brings in novelty, this time in terms of results.

Conclusions: Please add this section, together with perspectives of your study.

References do not follow the recommendations of MDPI journals, please check Guidelines for Authors and modify accordingly.

All these suggested changes should be performed in order to bring further improvements to the manuscript.

Author Response

Dear professor,

Thank you for your valuable comments, we are glad to respond to the comments in this file 

General comments

comment

Respond

Overall, in my opinion the topic of this research work is not suitable for “Metabolites”. At least, I would expect to see some analytical results regarding the chemical characterization of the Bee venom, and possible correlations between specific chemical constituents and the observed effects of treatment in animals.

Thank you for your comment, but due to multiple analytical results detected and its diversity in biochemical, histopathological and molecular ways, it was very difficult to collect it in the title, but instead it collected in refer to studying the effects on diabetic cardiomyopathy and nephropathy which belongs to the scope of "metabolites" 

Why the SZT-treated mice model was chosen? How can you assure that the results you obtained were not due to interactions between SZT and the administered nanoparticles and venom bee?

Although there are several methods for induction of diabetes in rats, but STZ administration still considered one of the effective and easiest models to be induced.

It was induced through a single I/P injection and the induction is confirmed through measuring the serum glucose concentration, and the treatment with nanoparticles and bee venom didn't started until 6 weeks after induction of diabetes, so the only effect present at this time is the effects resulted from induced hyperglycemia and the improvement of the results that can occurred resulted only from our treatments. 

Writing English should be carefully revised. Several errors were found throughout the manuscript

All manuscript is fully revised and several errors corrected

Other Comments

Site of comment

Comment

Revision

Abstract

The meaning of STZ should be reported

The full name is added

Abstract

Lines 20-28: in my opinion it is better to report more in detail the effects of treatment, instead of the deleterious effects of STZ

All abstract is revised and the effects of treatments is added

Introduction

Lines 88-93: recent reports regarding the use of SeNPs in diabetes should also be introduced, such as: doi.org/10.3390/molecules27175642

The recent report is added

Methods

Lines 122-25: The approval code of the animal trial should be reported

Approval code is added

Methods

Figure resolution must be improved. Also, the use of letters to indicate the significance of the results needs to be explained in the figure captures

All figure resolutions is improved, scale bar added and the letters is explained in figure captures

Round 2

Reviewer 3 Report

Dear Authors,

The present study evaluates the effect of Selenium nanoparticles and/or Bee venom against STZ-induced diabetic cardiomyopathy and nephropathy. The authors performed some of the suggested changes after the first round of review. Some changes of the manuscript should still be performed in order to improve its quality:

 Major changes

Abstract: The main purpose of the study is still not clear. Check the abstract for the recommended number of words in the recommendations for authors.

Introduction: Authors need to compare the purposes of their study with similar ones (they are not presented) and clarify the novelty. In the present form it is still not clear.

Discussions: Authors should state once again what the study brings in novelty, this time in terms of results. It is still not clear.

All these suggested changes should be performed in order to bring further improvements to the manuscript.

Author Response

Dear professor,

Thank you for your valuable comments, we are glad to respond to the comments in this file 

comment

Respond

Abstract: The main purpose of the study is still not clear. Check the abstract for the recommended number of words in the recommendations for authors.

The main purpose was more clarified. The recommended number of words is 200 word and the abstract words are 197

Introduction: Authors need to compare the purposes of their study with similar ones (they are not presented) and clarify the novelty. In the present form it is still not clear.

The previous studies clarified the SeNPs and BV effects were clarified and their comparison to our purpose was added. The novelty was cleared and all marked as a red color words. 

Discussions: Authors should state once again what the study brings in novelty, this time in terms of results. It is still not clear.

The novelty of results was cleared
